# Therapeutic Perspectives of HIV-Associated Chemokine Receptor (CCR5 and CXCR4) Antagonists in Carcinomas

**DOI:** 10.3390/ijms24010478

**Published:** 2022-12-28

**Authors:** Wilfredo Alejandro González-Arriagada, Isaac E. García, René Martínez-Flores, Sebastián Morales-Pison, Ricardo D. Coletta

**Affiliations:** 1Facultad de Odontología, Universidad de Los Andes, Santiago 7620086, Chile; 2Centro de Investigación e Innovación Biomédica (CIIB), Universidad de los Andes, Santiago 7620086, Chile; 3Patología Oral y Maxilofacial, Hospital El Carmen Luis Valentín Ferrada, Maipú 9251521, Chile; 4Laboratorio de Fisiología y Biofísica, Facultad de Odontología, Universidad de Valparaíso, Valparaíso 2360004, Chile; 5Centro de Investigación en Ciencias Odontológicas y Médicas, Universidad de Valparaíso, Valparaíso 2360004, Chile; 6Centro Interdisciplinario de Neurociencias de Valparaíso, Universidad de Valparaíso, Valparaíso 2381850, Chile; 7Unidad de Patología y Medicina Oral, Facultad de Odontología, Universidad Andres Bello, Viña del Mar 2531015, Chile; 8Centro de Oncología de Precisión (COP), Facultad de Medicina y Ciencias de la Salud, Universidad Mayor, Santiago 7560908, Chile; 9Department of Oral Diagnosis and Graduate Program in Oral Biology, Piracicaba Dental School, University of Campinas, Piracicaba 13414-903, SP, Brazil

**Keywords:** cancer therapy, immunotherapy, chemokines, oncology

## Abstract

The interaction between malignant cells and the tumor microenvironment is critical for tumor progression, and the chemokine ligand/receptor axes play a crucial role in this process. The CXCR4/CXCL12 and CCR5/CCL5 axes, both related to HIV, have been associated with the early (epithelial–mesenchymal transition and invasion) and late events (migration and metastasis) of cancer progression. In addition, these axes can also modulate the immune response against tumors. Thus, antagonists against the receptors of these axes have been proposed in cancer therapy. Although preclinical studies have shown promising results, clinical trials are needed to include these drugs in the oncological treatment protocols. New alternatives for these antagonists, such as dual CXCR4/CCR5 antagonists or combined therapy in association with immunotherapy, need to be studied in cancer therapy.

## 1. Introduction

Cancer involves multiple events, such as uncontrolled proliferation and DNA repair failures, that trigger genetic instability, invasion, migration, angiogenesis, and metastasis. These later events depend on the interactions between malignant cells and many cells belonging to the tumor microenvironment, leading to tumor progression [1]. These processes require the activation or inhibition of different cell signaling pathways mediated by cell surface receptors and their ligands for which the chemokine ligand/receptor axes play key roles [2]. According to their chemical structure, chemokines comprise four subtypes of cytokines (C, CC, CXC, and CX3C) that act as the ligand on one or more receptors. On the other hand, chemokine receptors CR, CCR, CXCR, and CX3CR are G protein-coupled receptors activated for one or more subtypes of chemokines [3].

In cancer, and other diseases, a chemokine receptor may activate a proinflammatory or anti-inflammatory pathway, a duality exploited by neoplastic cells to improve the capacity to (i) evade the immune system, (ii) degrade the extracellular matrix, and (iii) invade the neural or vascular compartment producing metastasis [3,4]. In this sense, our great interest is the role of CXCR4/CXCL12 and CCR5/CCL5 axes in the pathogenesis of epithelial malignancies, including lung [5], gastric [6], pancreatic [7], colorectal [8], breast [9], ovarian [10], prostatic [4], hepatocellular [11], and head and neck carcinomas [12,13], as well as adenocarcinomas [14] and non-epithelial cancers such as melanoma [15], multiple myeloma [16], and lymphomas [17].

Preclinical investigations (in vitro and in vivo) have proved the efficacy of HIV-related chemokine receptor (HIVrCR) antagonists for cancer treatment. Chemokine receptor antagonist (CRA) drugs such as maraviroc (CCR5 antagonist) or plerixafor (CXCR4 antagonist) have shown a role in the suppression of cancer cell proliferation, migration, and metastasis [18,19]. Thus, the CRA-promoted blockade of these axes arises as an alternative or complementary therapy to improve outcomes in tumors resistant to radiotherapy and chemotherapy in carcinomas [11,20,21,22,23,24,25]. In this context, we aim to conduct a scoping review of the literature to describe the results of promising CRA therapies for carcinomas in order to identify the drugs to be used in the future.

## 2. Materials and Methods

This study followed the PRISMA extensions for Scoping Reviews.

### 2.1. Inclusion and Exclusion Criteria

Eligibility criteria included the articles related to CRA therapy and carcinoma published from 2010 to September 2022 in English. Clinical trials and in vitro and in vivo studies were included. Exclusion criteria were articles that do not study the role of CCR antagonists as therapy, with a focus on other drugs, with other study designs, other publication types, and with a focus on other malignancies (sarcoma or lymphoproliferative). Unpublished clinical trials were not included.

### 2.2. Search Strategy

An electronic search was performed in the following databases: PubMed/Medline, Web of Science, and Scopus. The search terms were “chemokine receptor”, “CXCR4”, “CCR5” and “cancer therapy”.

PubMed/Medline: (“chemokine”(All Fields) AND “receptor”(All Fields)) OR (“chemokine receptor”(All Fields) OR “CXCR4”(All Fields) OR “CCR5”(All Fields)) AND (“cancer”(All Fields) AND “therapy”(All Fields)) OR (“cancer therapy”(All Fields)).

Web of Science: (((ALL = (chemokine receptor)) OR ALL = (CXCR4)) OR (ALL = (CCR5) AND ALL = (cancer therapy)).

Scopus: ABS (chemokine AND receptor OR cxcr4 OR ccr5 AND cancer AND therapy).

The references of the included studies and the relevant reviews were checked for possible further studies.

### 2.3. Study Selection

The title and abstract of all the articles were identified and selected using the web tool Rayyan (https://www.rayyan.ai/, accessed on 14 October 2022). The abstracts were read independently by the authors. Controversies were resolved by discussion between the authors.

### 2.4. Data Extraction

Data were tabulated using a specially designed form in Microsoft Excel^®^. The following data were extracted: author, country, year of publication, study design (three categories: clinical trial, in vitro and in vivo studies), receptor, antagonist, adjuvant therapy, type of cancer, oncogenic mechanisms involved, effective doses in vitro, effective doses in vivo, metastasis in vivo, survival in vivo and primary tumor size in vivo.

### 2.5. Data Analysis

A qualitative synthesis of the extracted data was performed through the resume table.

## 3. Results

### 3.1. Study Selection

A flowchart describing the selection process is presented in Figure 1. The search strategy returned 3807 articles published between 2010 and 2022 in all the databases. Duplicate records (1853 articles) were excluded, and 1954 remained for screening. After screening by title and abstract, the 150 remaining studies were assessed for eligibility with the full text. Finally, 89 articles were excluded based on the reasons detailed in Figure 1, and 61 publications were included (53 preclinical studies and 8 clinical trials).

### 3.2. Description of Preclinical Studies

Appendix A summarizes a qualitative synthesis of the preclinical studies. Forty-three publications reported CXCR4 antagonists, and ten reported CCR5 antagonists.

We found different inhibitors reported, AMD3100 (plerixafor) and maraviroc, the most frequent for CXCR4, and CCR5, respectively. Most studies included in vitro (16 studies) and in vivo (3 studies) methodologies.

Twenty-eight studies included adjuvant therapy. Among these therapies, the most reported were doxorubicin, gemcitabine, paclitaxel, and radiation. Breast cancer was the most common epithelial cancer included. The main oncogenic mechanisms these drugs interfered with were cell invasion, migration, and metastasis. However, the modulation of the immune response is another attractive action pathway of these drugs.

### 3.3. Description of Clinical Trials

A qualitative synthesis of the published clinical trials is summarized in Appendix A. All the papers were written in English and published between 2014 and 2022. Seven clinical trials reported CXCR4 antagonists (LY2510924, BL-8040, and mavorixafor), and two clinical trials were reported with a CCR5 antagonist (maraviroc). Three studies reported their results as monotherapy, and six studies used adjuvant therapy of durvalumab, pembrolizumab, nivolumab, sunitinib, and chemotherapy (leucovorin, liposomal irinotecan, and fluorouracil). The most reported inhibitor of CXCR4 was LY2510924, with a dose of 20 mg/day reported in different protocols. The inhibitor of CCR5, maraviroc, was used at 300 mg twice daily for two months. Other reported inhibitors of CXCR4 were motixafortide and mavorixafor. All the studies reported favorable survival results.

## 4. Discussion

The present review describes the current therapeutical perspectives of HIVrCR, CCR5, and CXCR4 antagonist drugs in treating carcinomas. Chemokines are associated with leukocyte chemoattraction to lymph nodes for maturation. However, cell migration of malignant keratinocytes, leukocytes, endothelial cells, mesenchymal stem cells (MSCs), and cancer-associated fibroblasts (CAFs) arise because of the aberrant expression of chemokine receptors on the surface of cancer cells [26]. Muller et al. were the first to demonstrate that the CXCR4/CXCL12 axis has a role in tumor progression and metastasis in breast cancer [27]. The interaction between cancer cells and the factors released by the microenvironment components is crucial for epithelial cancer progression, playing a role in the early and late events of cancer progression. The migration to lymph stimulated by oncochemoattractant depends on CCR5/CCL5 and CXCR4/CXCL12 axes [12]. The implantation of metastatic islands in lymph nodes requires the preparation of a premetastatic niche, a phenomenon related to the release of CCL5 and CXCL12 by the resident cells of lymph nodes [28,29]. CCL5 and CXCL12 have been linked to cancer progression, epithelial–mesenchymal transition, immune evasion, metastasis, and worse prognosis. Therefore, using antagonists such as small molecules (e.g., maraviroc or Vicriviroc), peptide antagonists, or antibodies have been suggested as therapeutic alternatives [26,30]. The small molecules have been tested in some clinical trials for carcinomas or other inflammatory diseases and are approved for HIV patients. Even if many chemokine receptors were essential for cancer progression, available drugs against HIVrCR have shown promissory results. Additionally, because these receptors are expressed in cancer and defensive cells, these inhibitors could halt cancer progression and modulate the immune response.

### 4.1. CXCR4/CXCL12 Axis

In normal tissues, this axis is important in developmental processes, hematopoiesis, and inflammation [31] and is expressed in leukocytes, stromal fibroblasts, and endothelial cells [32]. CXC chemokine receptor 4 (CXCR4) is a G-protein-coupled receptor highly expressed in different human carcinomas [4,6,7,8,11,13,29], at all stages of the epithelial–mesenchymal transition, invasion, or metastasis [33], and has been related to a poor prognosis. Leukocytes and CAFs are sources of CXCL12 [34], a chemokine that binds to CXCR4, forming the CXCL12/CXCR4 axis. This axis has a role in different tumor pathways involved in processes such as the epithelial–mesenchymal transition, cell migration, and metastasis, including drug resistance [35,36,37].

Small molecules, such as AMD3100 (plerixafor), WZ811, LFC131, AMD070, LY2510924, X4-136, BPRCX807, and others [19,38,39,40,41,42,43,44,45,46,47,48,49,50], have been reported to inhibit CXCR4, individually or in association with other drugs or therapies (doxorubicin, cisplatin, radiation, and others) [19,40,41,48,49,50,51,52], including different carriers for a better efficacy [10,33,52,53,54,55]. Preclinical studies have shown, both in vitro and in vivo, that the inhibition of CXCR4 is effective in treating cell proliferation, angiogenesis, tumor growth, and the metastasis of different carcinoma cells [19,25,38,39,40,41,42,43,44,51,52,53,54,56,57,58,59,60,61,62,63,64,65,66,67,68,69,70,71,72,73,74,75,76], and it has been reported that CXCR4 blockade increases tumor-infiltrating lymphocytes (TILs) [77]. These drugs modify cell migration and invasion and reduce metastasis [38,40,41,54]. Additionally, they have also been demonstrated to enhance the sensitivity to chemotherapy or radiotherapy, increasing the reduction in cell viability, apoptosis, tumor growth, and metastasis, including the modulation of the crosstalk between tumor and stromal cells [19,40,41,51,52,55,78]. Finally, plerixafor can induce a better immune response against the tumor through the suppression of Treg cells and the regulation of T cell activity [37,59], and recent studies have reported that CXCR4 inhibition enhances the response to immunotherapy [50,79]. Clinical trials have provided interesting data to consider CXCR4 antagonists as an alternative in cancer therapy in multicenter, randomized, and even phase II studies. Using these drugs as coadjuvant therapy has been successful, showing acceptable safety and tolerability in patients with advanced refractory tumors and expanding the benefits of chemotherapy or immunotherapy [47,80,81,82].

### 4.2. CCR5/CCL5 Axis

The use of CCR5-inhibitor drugs in HIV patients is well tolerated, and diverse clinical outcomes have been observed as monotherapy or combined with other antiretroviral drugs (highly active antiretroviral therapy; HAART). The association of CCR5 with cancer progression is unveiling a new perspective on the use of these drugs.

CC chemokine receptor 5 (CCR5) is a G-protein-coupled receptor reported in different kinds of carcinomas, with the primary role in the late events of cancer progression, such as metastasis [13,18]. CCL3 and CCL5 are the main chemokines that bind to CCR5, forming the CCL3/CCR5 and CCL5/CCR5 activation axes. These chemokines are mainly involved in inflammation, promoting the recruitment of leukocytes to injury sites [26]. The CCR5/CCL5 axis has protumor effects, and the low expression of these proteins can lead to a better prognosis [83].

In carcinomas, some authors have described the relationship between high levels of CCL5 or CCR5 expression in tumors and advanced stages [12,13,84,85,86], including a proangiogenic role [87] and the stimulation of cancer stem cells [88]. Regarding immune modulation, CCL5 can differentiate leukocytes to a protumorigenic profile [89] and can inhibit the antitumorigenic role of CD8+ lymphocytes [90] but can also modulate the activation of Tregs [18] and myeloid-derived suppressor cells [91]. CCL5 was reported as an inducer of cell migration and invasion [92], leading to metastasis [93]. The aggressiveness of CCL5-releasing tumors relies on the fact that they are more aggressive because CCL5 promotes invasion, migration, and metastasis in CCR5-high-expressing tumors.

CCR5 inhibition has demonstrated promising results in controlling cancer development and progression in preclinical studies [18,20,22,93,94,95,96,97]. Maraviroc is a specific small-molecule antagonist of the CCR5 used in preclinical and clinical cancer studies. CCR5 inhibitors were tested to treat liver, pancreatic, and breast cancer cells, showing apoptosis induction, reduced cell invasion and metastasis, and increased survival [20,93,98]. In addition, some studies reported that CCR5 inhibition could modulate the immune response, diminishing Treg infiltration [99,100].

It was reported that the inhibition of CCR5 in colorectal cancer cells, as a single agent, can inhibit proliferation and migration but failed to inhibit metastasis in vivo [101]. However, a study reported that maraviroc could inhibit metastasis in an animal model of colorectal cancer. These contradictory results are probably related to the promiscuity of chemokine receptors and chemokines, suggesting that drugs with dual or multiple inhibitions, or combined therapies (immunotherapy or chemotherapy), could have a better effect against cancer progression and metastasis. Recent preclinical studies have reported that the combination of CCR5 antagonists with anti-PD-L1 can inhibit tumor growth and enhance the therapy outcome in several types of cancer [100,102]. A few reported clinical trials are using CCR5 antagonists [103]. They are in phase I and use maraviroc.

### 4.3. New Perspectives

These chemokine axes play essential roles in cell migration and metastasis but can also modulate the immune response against tumors (Figure 2). Thus, studies about these axes as therapeutic targets are promising. The development of dual-inhibition drugs for CXCR4 and CCR5, and their association with immunotherapy, are the two main challenges in recent times. Therapies targeting CXCR4 and CCR5 have been reported as probable effective strategies for cancer. AMD3451 is the first reported dual CXCR4/CCR5 antagonist [104], but studies testing these drugs are necessary. Other drugs have also demonstrated a dual effect (e.g., NF279, penicillixanthone A, and GUT-70 [105]). The efficacy of these dual-inhibition drugs is still unknown in cancer.

A few reported clinical trials are testing these drugs in cancer, mainly in association with other therapies [45,46,47,106,107,108], and the results are still confusing. Hainsworth et al. reported a well-documented clinical trial with LY2510924 (anti-CXCR4) associated with sunitinib in metastatic renal cell carcinoma, showing promising results for this novel therapy [46]. On the other hand, clinical trials with maraviroc (antagonist of CCR5) and plerixafor (antagonist of CXCR4) have been reported, either as monotherapy or in combination with other drugs, but the literature mainly shows preclinical results in vitro or in vivo. Recent studies suggest that the chemokine receptor–ligand axis can modulate the tumor immune microenvironment [79]. Moreover, combining CXCR4 or CCR5 antagonists with anti-PD1 immunotherapy has shown promising results in colorectal cancer, renal cell carcinoma, and pancreatic ductal adenocarcinoma, including clinical trials [77,103,104,105,106,108]. These results reveal new alternatives to this therapy for human carcinomas.

## 5. Conclusions

The HIV-associated chemokine receptor (CCR5 and CXCR4) antagonists have critical roles in inhibiting cell migration, invasion, and metastasis and can modulate the tumoral immune response through Treg regulation. Preclinical studies are promising, but more clinical trials are needed. Future studies are required to test the efficacy of dual CXCR4/CCR5 antagonists and their association with immunotherapy.

## Figures and Tables

**Figure 1 ijms-24-00478-f001:**
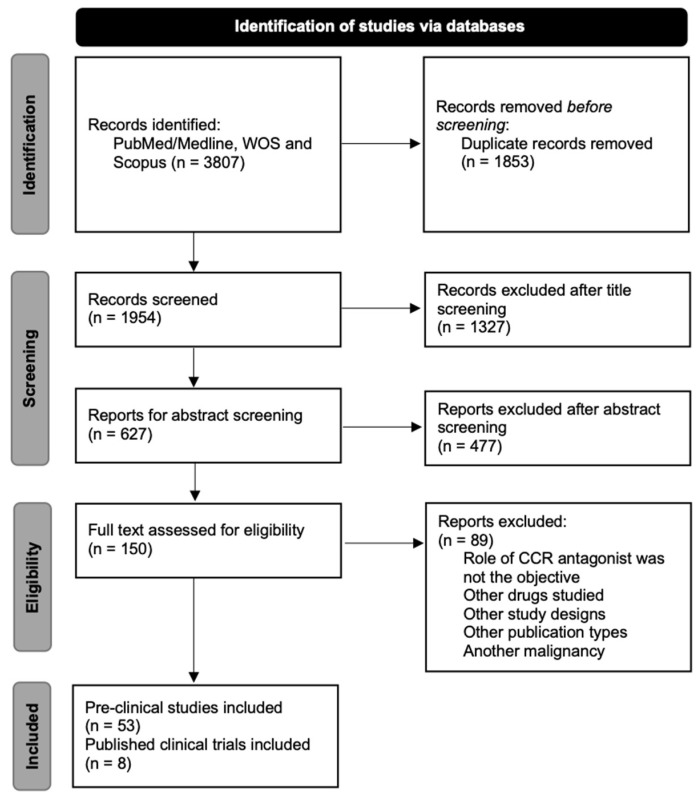
Flow diagram of the systematic selection of the studies.

**Figure 2 ijms-24-00478-f002:**
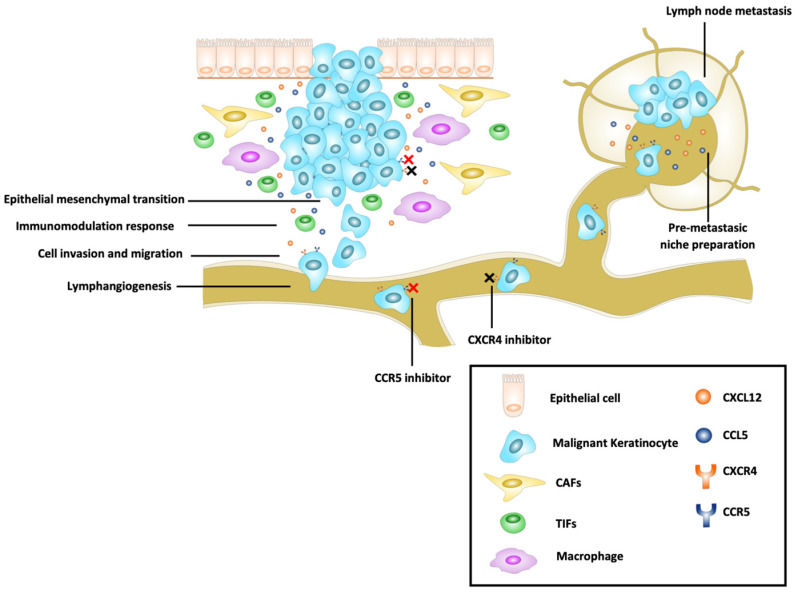
Schematic representation of oncogenic mechanisms mediated by the CCR5/CCL5 and the CXCR4/CXCL12 axes in carcinomas. The primary local sources of CCL5 and CXCL12 are CAFs, TILs, cells, macrophages, and tumor cells, while in lymph nodes, the primary sources are lymphatic fibroblasts, endothelial cells, lymphocytes, and tumor cells. These axes have a role in early events, such as epithelial–mesenchymal transition, local immune response modulation, local cell invasion, migration, and lymphangiogenesis, and in late events, such as premetastatic niche preparation, migration to lymph nodes (oncochemotaxis), and lymph node metastasis.

## Data Availability

Not applicable.

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
