# Peer review of "Therapeutic Perspectives of HIV-Associated Chemokine Receptor (CCR5 and CXCR4) Antagonists in Carcinomas"

_ijms, 2022, doi:10.3390/ijms24010478_

Round 1

Reviewer 1 Report

The manuscript indicates that chemokine ligand/receptor axes play key roles in the interaction between malignant cells and tumor microenvironment and summarizes the efficacy of dual CXCR4/CCR5 antagonists with 61 publications associated with CCR5 and CXCR4. This study helps to make clear their association with immunotherapy. Generally, the manuscript is well structured, and the contents are informative. Several format and grammatical issues should be addressedSpecific comment:

1. Line 45-50, according to your statement, chemokine ligand/receptor axes are essential in carcinomas. However, chemokines comprise four subtypes of cytokines, and their receptors can be activated by one or more subtypes of chemokines. Why do you choose CCR5 and CXCR4?

2. Line 125, “2 papers”.

3. Line 126, “6 studies”.

4. Line 139-141, the grammar of “Muller et al., studying breast cancer, were the first authors to describe that the CXCR4/CXCL12 axis has a role in tumor progression and metastasis” is not correct.

5. Line 164-167, long sentence, please divide into 2 sentences.

6. Line 181, “o T cell activity”? please modify.

7. Line 194, “downregulation of these proteins”, only gene expression can be downregulated, please modify.

8. Line 209, the grammar of “CCR5 inhibitors were tested to treat hepatocarcinoma, pancreatic cancer, and breast tumor cells, inducing apoptosis, reducing cell invasion and metastasis, and increasing survival” is not correct.

9. Line 213, “but failed to inhibit metastasis in vivo [99]. However”.

10. Line “a study reported recently shown that”, “reported,” and “shown” repeat, please modify. 

11. Line 225, the grammar of “Studies with these receptor antagonists have two main challenges nowadays, the dual inhibition of CXCR4 and CCR5 with only one drug, and the association with immunotherapy” is not correct.

12. Line 227-231, long sentence, please modify.

13. Line 235, use the abbreviation when it appears a second time. Please check and correct the full text.

14. Line 237, 239, what do “A” and “B” mean? Please modify.

15. The research lists a large number of advances in CCR5 and CXCR4, but the lack of its own summary and in-depth discussion makes the article lack coherence.

16. Many problems with reference format. The format of the page number and the paper abbreviations is not uniform. Please check throughout the reference.

Author Response

Reviewer #1

International Journal of Molecular Sciences,

We are pleased to respond to your questions and criticisms of the manuscript ijms-2051037. We appreciate the time and effort invested by you in examining our study. We have reviewed and answered all comments, that have helped to improve the quality of our manuscript. The alterations were made in the revised manuscript and highlighted in red. We hope that this new version meets the requirements for publication.

Thanks for your consideration.

The manuscript indicates that chemokine ligand/receptor axes play key roles in the interaction between malignant cells and tumor microenvironment and summarizes the efficacy of dual CXCR4/CCR5 antagonists with 61 publications associated with CCR5 and CXCR4. This study helps to make clear their association with immunotherapy. Generally, the manuscript is well structured, and the contents are informative. Several format and grammatical issues should be addressed. Specific comment:

  1. Line 45-50, according to your statement, chemokine ligand/receptor axes are essential in carcinomas. However, chemokines comprise four subtypes of cytokines, and their receptors can be activated by one or more subtypes of chemokines. Why do you choose CCR5 and CXCR4?

R:/ We choose these chemokine receptors because they are associated to HIV (Bleul et al., 1997), which is the focus of this manuscript.

(Ref. Bleul, C. C., Wu, L., Hoxie, J. A., Springer, T. A., & Mackay, C. R. (1997). The HIV coreceptors CXCR4 and CCR5 are differentially expressed and regulated on human T lymphocytes. Proceedings of the National Academy of Sciences, 94(5), 1925-1930.)

  1. Line 125, “2 papers”.

R:/ The line was modified and highlighted in red.

  1. Line 126, “6 studies”.

R:/ The line was modified and highlighted in red.

  1. Line 139-141, the grammar of “Muller et al., studying breast cancer, were the first authors to describe that the CXCR4/CXCL12 axis has a role in tumor progression and metastasis” is not correct.

R:/ The line was modified and highlighted in red.

  1. Line 164-167, long sentence, please divide into 2 sentences.

R:/ The sentences were modified and highlighted in red.

  1. Line 181, “o T cell activity”? please modify.

R:/ The line was modified and highlighted in red.

  1. Line 194, “downregulation of these proteins”, only gene expression can be downregulated, please modify.

R:/ The line was modified and highlighted in red.

  1. Line 209, the grammar of “CCR5 inhibitors were tested to treat hepatocarcinoma, pancreatic cancer, and breast tumor cells, inducing apoptosis, reducing cell invasion and metastasis, and increasing survival” is not correct.

R:/ The line was modified and highlighted in red.

  1. Line 213, “but failed to inhibit metastasis in vivo [99]. However”.

R:/ The line was modified and highlighted in red.

  1. Line 214 “a study reported recently shown that”, “reported,” and “shown” repeat, please modify.

R:/ The line was modified and highlighted in red.

  1. Line 225, the grammar of “Studies with these receptor antagonists have two main challenges nowadays, the dual inhibition of CXCR4 and CCR5 with only one drug, and the association with immunotherapy” is not correct.

R:/ The line was modified and highlighted

  1. Line 227-231, long sentence, please modify.

R:/ The line was modified and highlighted in red.

  1. Line 235, use the abbreviation when it appears a second time. Please check and correct the full text.

R:/ The line was modified and highlighted in red.

  1. Line 237, 239, what do “A” and “B” mean? Please modify.

R:/ The line was modified and highlighted in red.

  1. The research lists a large number of advances in CCR5 and CXCR4, but the lack of its own summary and in-depth discussion makes the article lack coherence.

R:/ Thanks for your comment, the summary was modified.

  1. Many problems with reference format. The format of the page number and the paper abbreviations is not uniform. Please check throughout the reference.

R:/ Thanks for your comment, the references were modified.

Reviewer 2 Report

The manuscript is a scoping review of the literature to describe the results of Chemokine Receptor Antagonists therapies for carcinomas  inlcuding both preclinical and clinical trials. The background is clear, the methods adequate and also the discussion is complete. However, it could be useful to better distinguish between preclinical and clinical trials when discussing the results. It is not clear what are the evidence and limitations of clinical trials in the Discussion

Author Response

Reviewer #2

International Journal of Molecular Sciences,

We are pleased to respond to your questions and criticisms of the manuscript ijms-2051037. We appreciate the time and effort invested by you in examining our study. We have reviewed and answered all comments, that have helped to improve the quality of our manuscript. The alterations were made in the revised manuscript and highlighted in red. We hope that this new version meets the requirements for publication.

Thanks for your consideration.

The manuscript is a scoping review of the literature to describe the results of Chemokine Receptor Antagonists therapies for carcinomas, including both preclinical and clinical trials. The background is clear, the methods adequate and also the discussion is complete. However, it could be useful to better distinguish between preclinical and clinical trials when discussing the results. It is not clear what are the evidence and limitations of clinical trials in the Discussion.

R:/ We agree with your comments. We have improved our discussion about clinical trials in the revised manuscript.

Reviewer 3 Report

The manuscript review the HIV-associated chemokine receptor (CCR5 and CXCR4) antagonists in carcinomas treatment. The authors mentioned the methods and result of search the articles related to CRA therapy and carcinoma published from 2010 to September 2022, but them are not necessary for Review. The authors introduce the CCR5/CCL5 and CXCR4/CXCL12 axes in cancer treatment, however, there is no detailed introduction to their current research status.

1. For Review, the “Materials and Methods” and “Results” sections are not necessary, the "Discussion" section can be broken down into sections summarizing.

2. What is the principle of HIV-associated chemokine receptor antagonists? How it uses the tumor microenvironment to exert its therapeutic effect?

3. Compared with other treatment methods currently used clinically, what are the advantages of HIV-associated chemokine receptor antagonists?

4. What is the current status of clinical application of HIV-associated chemokine receptor antagonists, and what are the limitations of clinical application?

5. Please provide detailed examples of current research on CCR5 and CXCR4 to show the current status of research on them.

Author Response

Reviewer #3

International Journal of Molecular Sciences,

We are pleased to respond to your questions and criticisms of the manuscript ijms-2051037. We appreciate the time and effort invested by you in examining our study. We have reviewed and answered all comments, that have helped to improve the quality of our manuscript. The alterations were made in the revised manuscript and highlighted in red. We hope that this new version meets the requirements for publication.

Thanks for your consideration.

The manuscript review the HIV-associated chemokine receptor (CCR5 and CXCR4) antagonists in carcinomas treatment. The authors mentioned the methods and result of search the articles related to CRA therapy and carcinoma published from 2010 to September 2022, but them are not necessary for Review. The authors introduce the CCR5/CCL5 and CXCR4/CXCL12 axes in cancer treatment, however, there is no detailed introduction to their current research status.

  1. For Review, the “Materials and Methods” and “Results” sections are not necessary, the "Discussion" section can be broken down into sections summarizing.

R:/ Thanks for your comment. We think that for a scoping review, a systematization of the bibliographic search is necessary, and that the presentation of results is better keeping these sections.

  1. What is the principle of HIV-associated chemokine receptor antagonists? How it uses the tumor microenvironment to exert its therapeutic effect?

R:/ A chemokine antagonist block the binding site of chemokines to their specific receptor, inhibiting the signaling pathways triggered by their binding (Miao et al., 2020). The action of chemokines has an impact on tumor cells and its microenvironment because they are released by inflammatory cells or endothelium, inducing epithelial mesenchymal transition, invasion, migration and consequently tumor progression (Kraus et al., 2021).

  • Miao, M., De Clercq, E., & Li, G. (2020). Clinical significance of chemokine receptor antagonists. Expert Opinion on Drug Metabolism & Toxicology, 16(1), 11-30.
  • Kraus, S., Kolman, T., Yeung, A., & Deming, D. (2021). Chemokine receptor antagonists: Role in oncology. Current Oncology Reports, 23(11), 1-10.

  1. Compared with other treatment methods currently used clinically, what are the advantages of HIV-associated chemokine receptor antagonists?

R:/ Chemokines modulate the tumoral microenvironment and cancer progression, in different ways, and the drugs currently used for cancer therapy, including immune checkpoint inhibitors are still insufficient to obtain a complete remission of cancer. We think that the benefits of a combined therapy, including HIV-associated chemokine receptor antagonists, in patients with a positive expression of these receptors in the tumor, could improve the currently reported results with chemotherapy, immune checkpoint inhibitors or other proposed target therapies.

  1. What is the current status of clinical application of HIV-associated chemokine receptor antagonists, and what are the limitations of clinical application?

R:/ Both antagonists were developed for HIV-treatment. The current status of Plerixafor, the most common CXCR4 antagonist, is for his action on hematopoietic stem cells for treatment of lymphoproliferative neoplasia. Maraviroc, is the most common CCR5 antagonist and is currently used for the treatment of HIV patients (Miao et al., 2020). In oncology, both antagonists are recently used in phase I and phase II clinical trials and are not included in standard protocols for solid tumors.

  • Miao, M., De Clercq, E., & Li, G. (2020). Clinical significance of chemokine receptor antagonists. Expert Opinion on Drug Metabolism & Toxicology, 16(1), 11-30.

  1. Please provide detailed examples of current research on CCR5 and CXCR4 to show the current status of research on them.

R:/ Current research about CCR5 and CXCR4 in oncology is detailed in both supplementary tables.

Round 2

Reviewer 1 Report

The revised version is accepted 

Author Response

Thanks for your comments.

Reviewer 3 Report

This manuscript has been improved.

Author Response

Thanks for your comments and the English language will be professional proofreading.